# *Phytophthora* zoospores display klinokinetic behaviour in response to a chemoattractant

**Michiel Kasteel**[1,2], **Tharun P. Rajamuthu**[1,2¤], **Joris Sprakel**[3☼], **Tijs Ketelaar**[1☼]*, **Francine Govers**[2☼]

1 Laboratory of Cell and Developmental Biology, Wageningen University & Research, Wageningen, the Netherlands, 2 Laboratory of Phytopathology, Wageningen University & Research, Wageningen, the Netherlands, 3 Laboratory of Biochemistry, Wageningen University & Research, Wageningen, the Netherlands

☼ These authors contributed equally to this work.
¤ Current address: Bayer Crop Science, Hoofddorp, the Netherlands
* tijs.ketelaar@wur.nl

**Data Availability Statement:** All coding is available from https://zenodo.org/records/11632432. Data is available from https://zenodo.org/records/13353473.

## Abstract

Microswimmers are single-celled bodies powered by flagella. Typical examples are zoospores, dispersal agents of oomycete plant pathogens that are used to track down hosts and infect. Being motile, zoospores presumably identify infection sites using chemical cues such as sugars, alcohols and amino acids. With high-speed cameras we traced swimming trajectories of *Phytophthora* zoospores over time and quantified key trajectory parameters to investigate chemotactic responses. Zoospores adapt their native run-and-tumble swimming patterns in response to the amino acid glutamic acid by increasing the rate at which they turn. Simulations predict that tuneable tumble frequencies are sufficient to explain zoospore aggregation, implying positive klinokinesis. Zoospores thus exploit a retention strategy to remain at the plant surface once arriving there. Interference of G-protein mediated signalling affects swimming behaviour. Zoospores of a *Phytophthora infestans* Gα-deficient mutant show higher tumbling frequencies but still respond and adapt to glutamic acid, suggesting chemoreception to be intact.

## Author summary

Oomycetes of the genus *Phytophthora* are devastating plant pathogens that have enormous impact on world food production. Unlike most of the morphologically similar Fungi, *Phytophthora* species make use of motile dispersal agents called zoospores. Although it is known that zoospores end up at specific parts of the plant, it is unknown in what manner their motility contributes to this result. A better understanding of how zoospores respond to environmental signals, for example to exudates deriving from plants, will open novel avenues to combat the diseases caused by *Phytophthora*. We exploited an imaging system that allows us to directly monitor zoospore trajectories and have used this to observe and quantify changes in zoospore behaviour in response to a plant-exuded compound. Doing so, we found that zoospores have access to a particular type of aggregation strategy which they use to target the plant. Moreover, we were able to distinguish a

**Funding:** This publication is part of the project 'Blight in the spotlight; an innovative microscopic assay for unravelling and quantifying the Phytophthora infestans infection process' (project number GSGT. GSGT.2018.024) of the research programme Graduate School Green Top Sectors which is financed by the NWO Science domain (NWO-ENW, https://www.nwo.nl/en/science-enw) of the Dutch Research Council (NWO), awarded to Michiel Kasteel (MK). The funder did not have a role in either the study design, data collection and analysis, decision to publish, or preparation of the manuscript.

**Competing interests:** The authors have declared that no competing interests exist.

zoospore's capacity to sense a signal from its capacity to act on that signal, a criterium that is essential for unravelling the mode of action of crop protection agents.

## Introduction

Microswimmers are single-celled bodies powered by one or more flagella and widespread across the tree of life. In aqueous environments they exploit their motility to find mating partners or sustenance. To do so, they must be able to sense and interpret environmental signals to steer towards their goal. Due to their relatively small size, many microswimmers are considered unable to spatially recognize signal gradients, and must discriminate gradient direction by scanning their environment [1]. One of the most prominent search strategies applied by microswimmers, and extensively studied in the bacterium *Escherichia coli* [2], is the run-and-tumble motion, where straight swimming stretches are alternated by rapid turns. Adapting their tumbling frequency to increasing concentrations of the attractant allows these bacteria to move up the gradient, a process termed klinokinesis [3]. Run-and-tumble behaviour is not only omnipresent in prokaryotes, eukaryotic microswimmers such as the alga *Chlamydomonas reinhardtii* employ this type of motility as well [4].

Oomycetes of the genus *Phytophthora* can cause devastating diseases on a wide variety of crops and considerable damage in forests and natural ecosystems. Notorious species are *Phytophthora infestans* on potato [5], *Phytophthora palmivora* on cocoa and oil palm [6], *Phytophthora capsici* on peppers and cucurbits [7], and *Phytophthora sojae* on soybean [8]. Under conducive circumstances they can nullify crop yields [9] and even with intensive pesticide spraying regimes [10], damage is still substantial. *P. infestans* for example, remains responsible for destroying 15–20% of global potato yield [11]. The ever-heavy disease pressure of *Phytophthora* is often attributed to its short asexual reproduction cycle, with a massive release of dispersal agents, i.e. spores [12]. As opposed to most of their morphologically similar but phylogenetically distinct fungal counterparts, oomycetes also produce motile dispersal agents termed zoospores [13]. These wall-less, flagellated mononuclear cells are typical examples of microswimmers. As outlined in a recent review [14] *Phytophthora* zoospores show a multitude of taxes including chemotaxis and are thought to use their motility and sensing capacity to propel themselves through saturated soils and water films on leaves to actively target their host. This behaviour known as 'homing' [14], is evident from their localized accumulation on plant roots [15], stomates [16,17] and wounds [18]. Zoospore aggregation is induced by a variety of plant derived components, such as amino acids, sugars, alcohols and flavonoids in root exudates. An example of such a root exudate component is the amino acid glutamic acid (Glu), which induces zoospore aggregation not only in all studied plant pathogenic oomycetes [14], but even in the oomycete fish pathogen *Saprolegnia diclina* [19].

In eukaryotes G-protein mediated signalling plays a crucial role in sensing environmental signals. In the canonical G-protein signalling pathway membrane-spanning G-protein-coupled receptors (GPCRs) are activated by a very diverse range of ligands such as, for example, neurotransmitters, pheromones and volatiles. This activation triggers dissociation of the heterotrimeric G-protein complex releasing the Gα subunit from Gβγ as initiators for further downstream signalling. Zoospores of *P. infestans* and *P. sojae* transformants lacking the Gα subunit show aberrant swimming patterns and do not aggregate at Glu sources [20,21]. Moreover, the *P. sojae* transformants no longer aggregate to daidzein, an isoflavone secreted by soybean in the rhizosphere as sensor for beneficial microbes but hijacked by pathogens [21]. Likewise, *P. sojae* transformants in which a non-canonical GPCR gene is silenced show defects in zoospore swimming behaviour and aggregation suggesting a role for this GPCR as

chemoreceptor [21,22]. Notably, besides canonical GPCRs oomycetes have a remarkable repertoire of unique non-canonical GPCRs but as yet their functions are unknown [23].

Although zoospores have been described to actively approach attractant sources [24], the manner in which they do so remains unclear. In a recent review [14], we summarized studies on aggregation and behaviour of zoospores in response to chemoattractants. We noted that earlier reports on zoospore behaviour are limited to descriptions of observations, occasionally supported by hand-drawn tracings [25–27] or long exposure-images [27] of zoospore trajectories, using terms such as 'excited' [24], 'jerky' [28] and 'milling' [29] to define a zoospore's response to attractants. It appears that various authors use different terminology to describe the same behavioural sequence displayed by zoospores. In an attempt to harmonize the terminology we coined the following terms for the successive stages in the homing response: (i) reorientation, (ii) approaching, (iii) retention and (iv) settling. Recent studies exploited more advanced tools for direct visualization of zoospore trajectories, allowing for increased quantification and precise characterization of swimming behaviour. As a result, meticulous analyses of *Phytophthora parasitica* zoospores [30] enabled identification of the parameters controlling the distinct stages of the zoospore homing response [31,32].

In this study, we design a setup for high-speed microscopy of microswimmers and use this to track *Phytophthora* zoospores over time when exposed to the chemoattractant Glu or the non-attractant amino acid phenylalanine (Phe). We show that in the absence of an attractant, zoospores swim in a run-and-tumble pattern, where straight swimming stretches are alternated by rapid turns. In the presence of Glu, we find that the tumbling frequency increases in a dose-dependent manner while the swimming velocity and flagellar beating frequency are unaltered. We show that the increased tumbling frequency is sufficient to explain chemotaxis through stochastic aggregation. Zoospores produced by a *P. infestans* Gα subunit mutant [20] show an increased tumbling frequency even in the absence of an attractant, but still respond to Glu in the same fashion as the wildtype does.

## Results

### *Phytophthora* zoospores run and tumble

To investigate *Phytophthora* zoospore swimming, we use high spatio-temporal resolution microscopy to track zoospore movement over time. Four different species were examined, i.e., *P. palmivora*, *P. sojae*, *P. capsici* and *P. infestans*. From the obtained trajectories (Figs 1A and S1), we extract parameters characterizing zoospore behaviour, i.e., velocity $v$, curvature $k$ and moving direction $\theta$ (Fig 1A). The results show that zoospores from all four species perform straight runs typically lasting less than a second (S2 Fig) before being interrupted by reorientations through tumbling. These tumbles typically occur within a second (S2 Fig) and, as shown for *P. infestans* zoospores, through flagellar desynchronization (S3 Fig). Tumbling occurs at a wide range of angles anywhere between 30˚ and 180˚ (S4 Fig). Velocity plummets during tumbling, after which it recovers to a constant run velocity (Fig 1A). This suggests a zoospore's average velocity (S2 Fig) to be dictated by tumble frequency rather than adaptable running velocity. The difference in basal velocity and tumbling frequencies between *Phytophthora* species dictates how efficiently they explore, a parameter we express as active diffusion. High-velocity, low-tumbling *P. sojae* zoospores show a much higher basal active diffusion than low-velocity, high-tumbling *P. infestans* zoospores (Fig 2B).

### Glutamic acid reduces active diffusion by increasing tumble frequency

To investigate how attractants influence swimming behaviour, zoospores were exposed to increasing concentrations of Glu covering the range that has been shown to induce zoospore

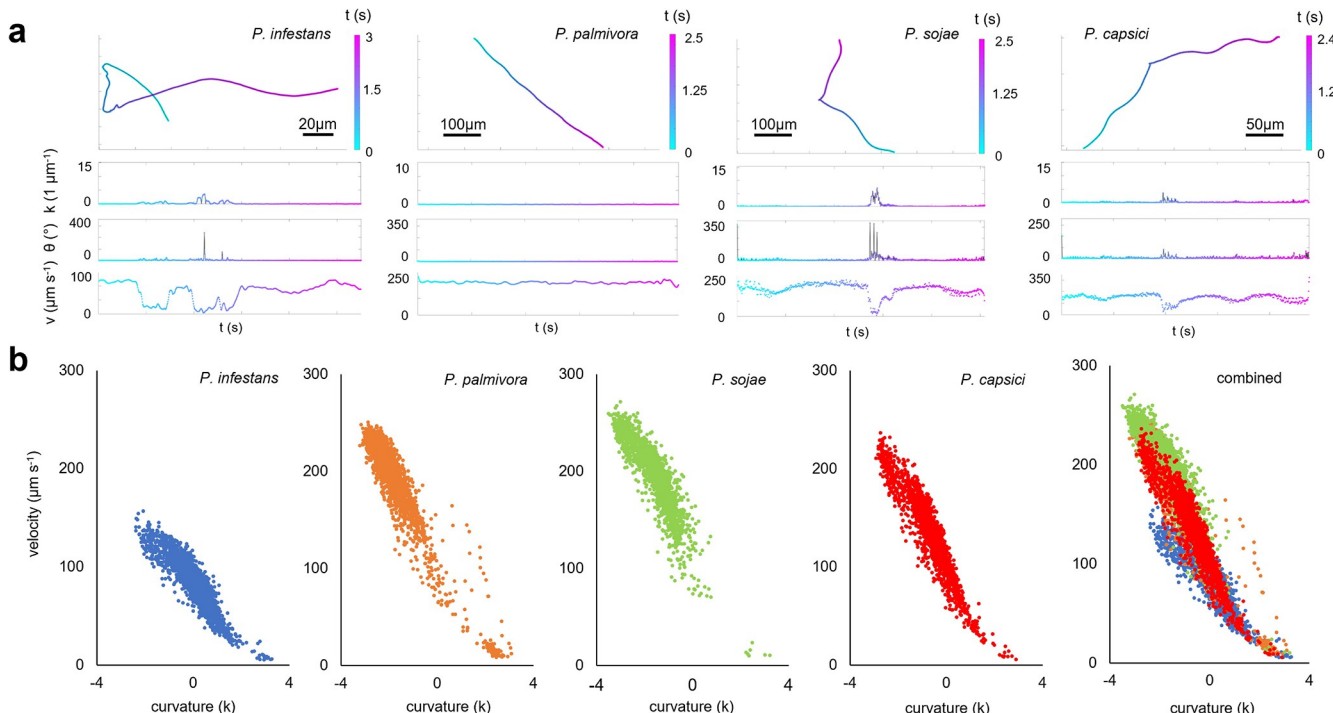

**Fig 1. Swimming trajectories of *Phytophthora* zoospores.** Trajectories are deduced from high-speed microscopy images of *P. infestans*, *P. palmivora*, *P. sojae* and *P. capsici* zoospores swimming in water in microchambers. Minimum number of zoospores imaged per species is 3251. (a) Representative trajectories. Curvature ($k$) in 1 µm⁻¹, moving direction ($\theta$) in ° and velocity ($v$) in µm s⁻¹ are plotted over time (t) in seconds (s). (b) Kinetic fingerprints of swimming zoospores in which each dot depicts the mean velocity for a trajectory of a single zoospore (Y axis) and the logarithm of the mean curvature for that trajectory (X axis). On the right an overlay of kinetic fingerprints of the four species.

aggregation in several *Phytophthora* species [20,33–37]. Phe, an amino acid that has no activity as attractant [38], was included as a control. As expected, zoospores did not respond to Phe but did do so to Glu in the millimolar range (Figs 2A and S1). In the presence of Phe we did not find obvious changes in swimming trajectories or run velocities even at the highest tested Phe concentration (S1 and S5 Figs). Overall, in the presence of Glu average velocities dropped drastically (S2 Fig) but run velocities (Fig 2D) and flagellar beating patterns (S6 Fig) were unaffected. Instead, reduction of average velocity was solely affected through tumbles; Glu decreased run duration (S2 Fig), increased tumble duration (S2 Fig) and, at the highest tested Glu concentrations, induced a semi-permanent state of tumbling (Fig 2D). Increased tumbling times did not seem to correlate with increased tumbling angles (S4 Fig). As suggested from the velocity distributions, we find that the diversity of swimming patterns found in control settings (no Glu) is diminished in the presence of Glu, committing all zoospores to an increased tumbling state (Figs 2C and S5B). At the highest Glu concentrations, this even induces a new population, where a semi-permanent state of tumbling induces extremely low average velocities (Fig 2C), all resulting in a drastic decrease of active diffusion (Fig 2D).

## Tumble tuning is sufficient to induce aggregation at a source

Adapting tumbling frequencies in response to an attractant is reminiscent of klinokinesis, the mechanism bacteria use to move up a gradient. To investigate whether the adaptative tumbling frequencies that we observe are sufficient to explain aggregation at a source, we developed a simulation model of non-interacting run-and-tumble swimmers using input parameters (run

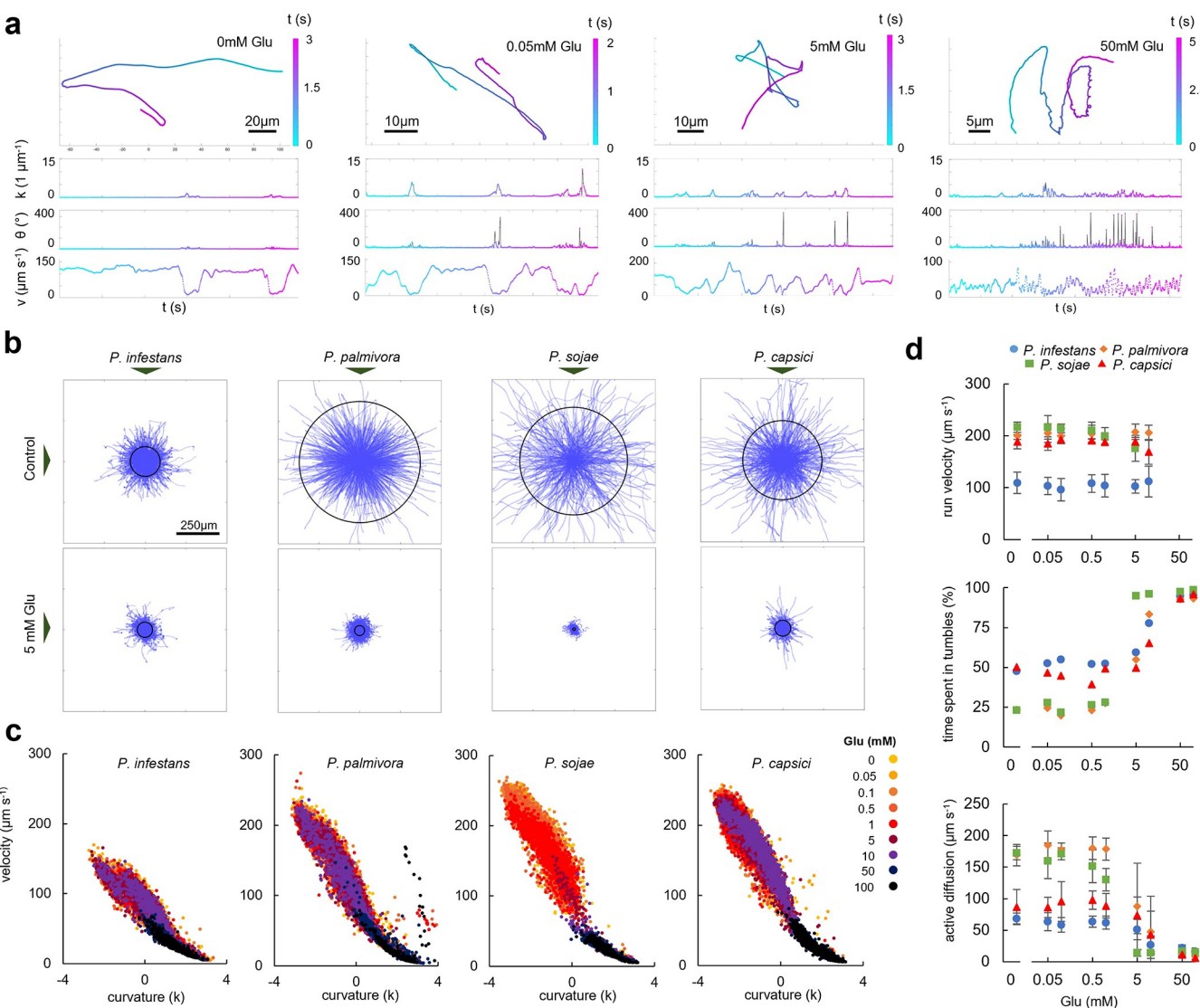

**Fig 2. Swimming trajectories of *Phytophthora* zoospores in response to glutamic acid (Glu).** Minimum number of zoospores imaged per treatment is 1672. (a) Representative trajectories of *P*. infestans zoospores swimming in microchambers in the absence (0) or presence of Glu at 0.05, 0.5 and 50 mM. Curvature ($k$) in 1 μm$^{-1}$, moving direction ($\theta$) in ° and velocity ($v$) in μm s$^{-1}$ are plotted over time (t) in seconds (s). (b) Trajectories of zoospores of four *Phytophthora* species in the absence of Glu (top) and in 5 mM Glu (bottom) depicted as overlays with the starting point of each trajectory in the centre. Black circles indicate the average net distance covered (active diffusion) by zoospores in 1 second. (c) Kinetic fingerprints of swimming zoospores in which each dot depicts the mean velocity for a trajectory of a single zoospore (Y axis) and the logarithm of the mean curvature for that trajectory (X axis) in the absence (0) or presence of Glu in concentrations up to 100 mM. (d) Effect of different concentrations of Glu (x-axis) on run velocity (top), percentage of the time spent in tumbles (middle) and active diffusion (bottom).

velocity and tumble rate) estimated from our experiments on *P. infestans* zoospores responding to Glu. We model an attractant source in the centre of a square simulation box which diffuses according to root exudate profiles described by Fickian diffusion [39] (S7 Fig), and trajectories of randomly distributed zoospores (Fig 3). With increasing source concentrations (S7B Fig), we find zoospores to increasingly aggregate at the source, showing that our measured increase in tumbling frequency in response to a chemoattractant is sufficient to explain aggregation at a source. *Phytophthora* thus has access to a positive klinokinetic aggregation strategy to target its host.

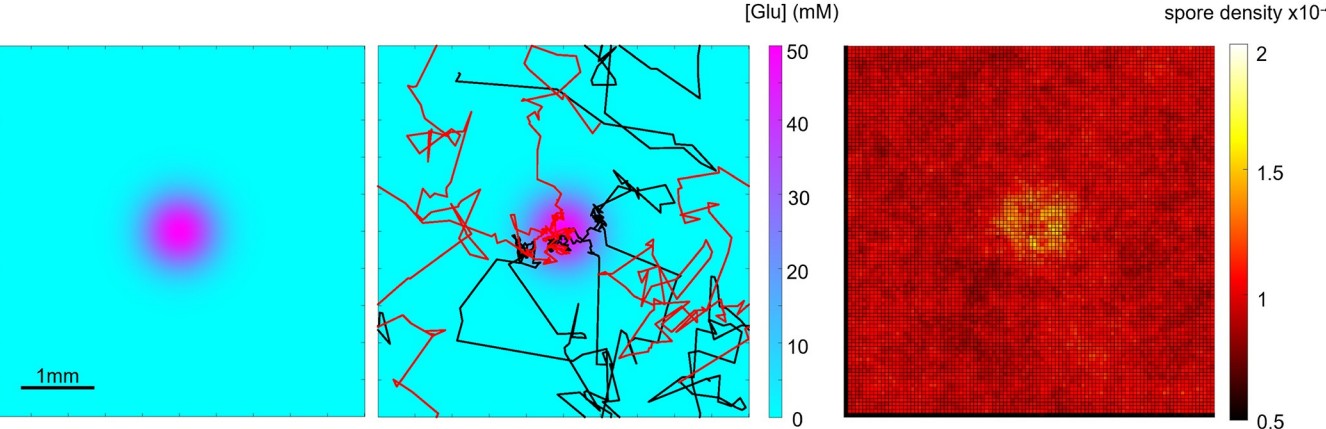

**Fig 3. Simulation model of zoospores reacting to a diffusive chemoattractant.** (left) A chemoattractant diffuses from a point source of 50 mM in a square chamber. (middle) Zoospore trajectories in response to the chemoattractant. Simulated trajectories of two individual zoospores (red and black) are plotted using our observed parameters on run velocity and adaptation of tumbling frequency in response to glutamic acid. (right) Predicted zoospore density after a 1000 zoospores were allowed to 'swim' for 1 hour. Zoospore density given as a time fraction of zoospore presence.

## Disabling the Gα subunit affects swimming behaviour but not Glu perception

Based on previous studies in *P. infestans* and *P. sojae* showing that silencing the Gα subunit gene results in mutants with defects in zoospore swimming behavior and chemotactic responses towards Glu [20,21], it was hypothesized that Glu perception is mediated by a GPCR that functions as chemoreceptor. An alternative hypothesis, however, is that this inability to swim towards the chemoattractant Glu is due to a defect in controlling swimming direction. To investigate this we used the high speed camera setup to analyze zoospore movement of one of the *Phytophthora* Gα mutants in more detail and in absence and presence of Glu. *P. infestans* gs2 is a transformant in which the Gα-subunit gene *Pigpa1* is silenced and PGA1 is not detectable [20]. In control settings (no Glu), we found gs2 zoospores to indeed show strongly aberrant swimming patterns as compared to the wildtype (Fig 4A), exhibiting a higher tumbling frequency and a severely decreased active diffusion (Fig 4B and 4D). Although flagellar beating patterns did not seem to be affected, Fourier analysis of of trajectory velocity fluctuations did indicate gs2 to tumble regularly, at about 1.4 Hz, whereas no such tumble regularity was found in Fourier analysis of wildtype zoospores (S6B Fig). When exposed to Glu, gs2 zoospores respond in the same manner as wildtype zoospores: Their tumbling frequency (Fig 4D) and mean velocity (S2 Fig) are decreased, while run velocity (Fig 4D) is not affected. Glu does not decrease active diffusion of gs2 zoospores. Rather, in the gs2 mutant active diffusion without Glu is already severely reduced, comparable to the levels displayed by the wildtype when exposed to high concentrations of Glu (Fig 4D). This shows that the loss of chemotaxis to Glu in gs2 zoospores is not due to a defect in Glu perception but likely to alterations in their swimming patterns, where increased and regular tumbles drastically decrease their basal active diffusion (Fig 4D).

## Discussion

*Phytophthora* zoospore aggregation at chemoattractant sources is well documented, but reports on how they accomplish this address qualitative rather than quantitative parameters [14]. Tran et al. [30] showed how high-speed microscopy allows for direct and quantitative monitoring of zoospore behaviour. In this study we use this tool to investigate zoospore

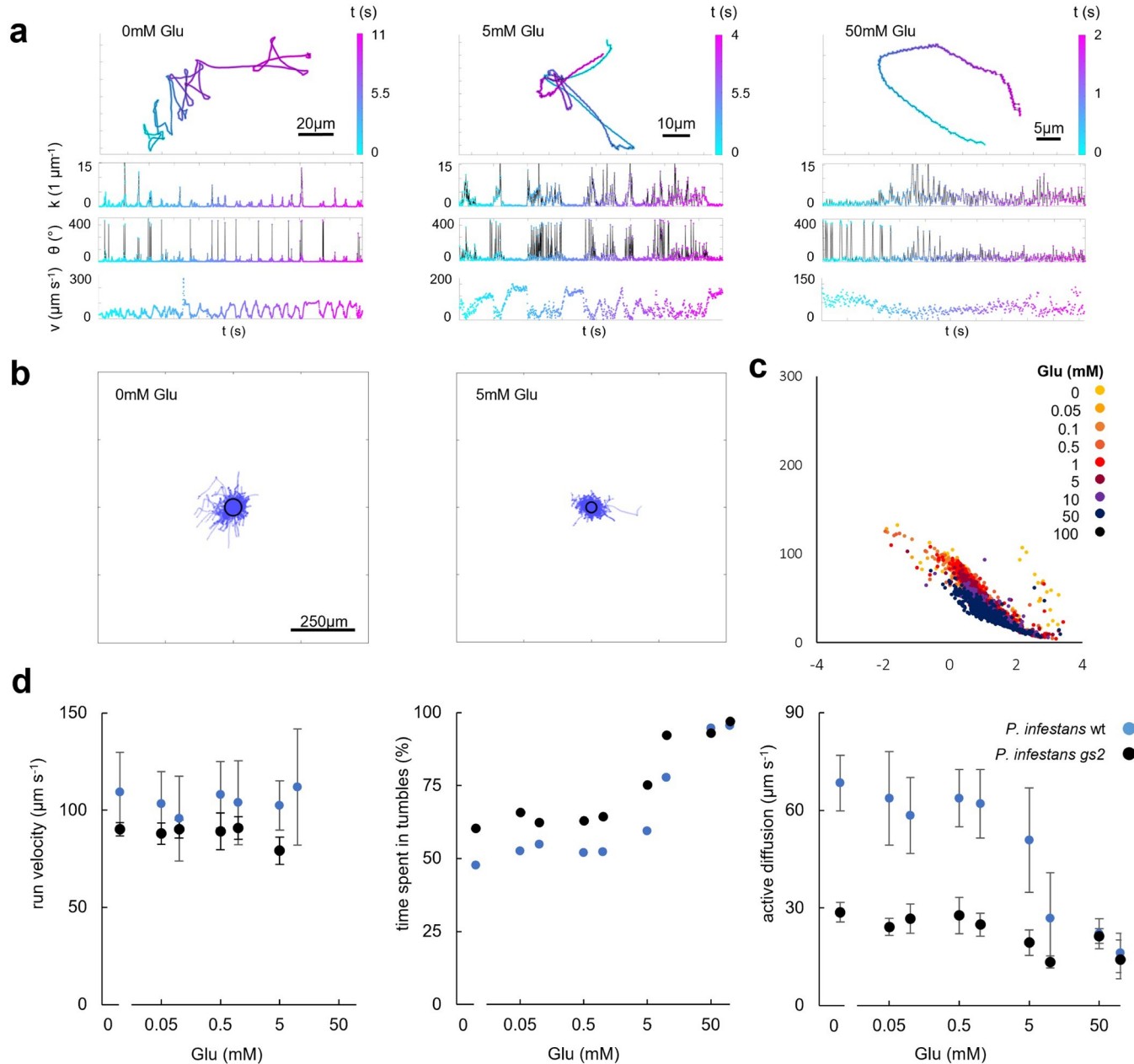

**Fig 4. Swimming trajectories of *P. infestans* Gα-silenced (gs2) zoospores in response to glutamic acid (Glu).** Minimum number of zoospores imaged per treatment is 219. (a) Representative trajectories of gs2 zoospores swimming in microchambers at 0, 0.05, 0.5 and 50 mM Glu. Curvature ($k$) in 1 μm$^{-1}$, moving direction ($\theta$) in ° and velocity ($v$) in μm s$^{-1}$ are plotted over time ($t$) in seconds (s). (b) Zoospore trajectories in the absence of Glu (left) and in 5 mM Glu (right) depicted as overlays with the starting point of each trajectory in the centre. Black circles indicate the average effective distance travelled (active diffusion) by zoospores in 1 second. (c) Overlay of kinetic fingerprints in which each dot depicts the mean velocity for a trajectory of a single zoospore (Y axis) and the logarithm of the mean curvature for that trajectory (X axis) in the absence of Glu (0) and in the presence of Glu in concentrations up to 100 mM. (d) Effect of Glu on run velocity (left), percentage of time spent in tumbles (middle) and active diffusion (right).

behaviour in multiple *Phytophthora* species. We also monitor and quantify changes in their swimming behaviour in response to a chemoattractant and investigate how a *P. infestans* transformant impaired in G-protein signaling [20] is affected in those aspects.

We found zoospores of *P. infestans*, *P. palmivora*, *P. capsici* and *P. sojae* to swim in patterns of straight stretches alternated by sudden changes of direction (Fig 2). This behaviour is known as run-and-tumble swimming, which Tran et al. [30] likewise ascribed to *P. parasitica* zoospores. Earlier work on *Phytophthora* zoospore behaviour also recognizes the alternation between runs and tumbles: our findings are in line with reports on *P. infestans* [20,25] and *P. palmivora* [40], and similar observations have been reported for *Phytophthora megasperma* [41] and *Phytophthora cinnamomi* [29,42]. Why would zoospores adapt such swimming patterns? Native run-and-tumbling (i.e., in the absence of attractants) is used by bacteria to pick up on host-derived signals [43]. Through run-and-tumble swimming, bacteria adopt exploration patterns called Lévy walks, allowing them to identify scarce nutrient sources [43]. Alternatively, run-and-tumbling could improve soil navigation: bacteria need to tumble in order to prevent getting stuck while navigating complex geometries [44]. Tumbling to prevent getting stuck might also allow oomycete zoospores to successfully migrate through the complex geometry of the soil [44–46], as exemplified by a zoospore's tendency to tumble upon collision with a surface [29].

Klinokinesis is a stochastic navigation mechanism whereby attractant concentrations determine tumbling frequency, thereby biasing the movement of microswimmers up the gradient. Bacteria typically perform negative klinokinesis, where attractants decrease tumbling frequency thereby promoting straight stretches toward the source [2]. In contrast, we found *Phytophthora* zoospores to increase tumble frequencies in response to increasing concentrations of the chemoattractant Glu (Fig 2D). This suggests that zoospores use positive rather than negative klinokinesis to aggregate, a mechanism that according to our simulation models should be sufficient for aggregation of zoospores at a point source. Positive klinokinesis was likely also observed by others in earlier experiments, but not coined as such. For example, *Phytophthora* and *Pythium* zoospores have been known to increase their tumbling frequency when nearing attractant sources, as well as in biological settings, i.e. when zoospores approach roots (as reviewed in [14]). Often, the process is described as 'trapping', with zoospores initially being drawn in by directed movement toward the source [47], a behaviour also reported in brown algae when attracted by sex pheromones [48,49].

We found zoospores to respond to Glu in the mM-range. How likely is a zoospore to encounter such concentrations in the phytosphere? The microbiome ensures that most plant exudates diffuse not further than only a couple of millimetres from the root [39], and it is thus likely that zoospores only perceive plant-derived glutamic acid (and other attractive primary metabolites) at what is basically the surface of plant wounds and root tips [18]. This arrival might be by pure chance or alternatively, by compounds suited to long-range attraction (like isoflavonoids [50]). Irrespective of their mode of arrival, we show how Glu can retain zoospores at the plant surface: Positive klinokinesis ensures zoospore net movement is greatly reduced, and thus results in aggregation at the site of interest. We therefore suggest that at least some components of the plant exudate, rather than inducing directional swimming, induce zoospore aggregation by retention.

The majority of the currently known receptors that govern chemotaxis are GPCRs, a large family of very diverse seven transmembrane spanning receptors that are widespread in eukaryotes. They act as gateways; upon sensing extracellular signals they activate heterotrimeric G-proteins and downstream intracellular signalling cascades to accomplish the desired response. G-protein mediated signalling has been shown to be involved in chemotaxis in *P. infestans* and *P. sojae* [20,21]. Our findings show that zoospores of a Gα-deficient *P. infestans* mutant that have lost the ability to aggregate and to swim toward Glu, show higher tumbling rates

(Fig 4D). It appears however, that they are still capable to sense Glu: they respond in the same manner as zoospores of the wildtype, and hence their capacity to sense and recognize Glu is not affected (Fig 4D). Lack of chemotaxis [20,21] therefore seems to be due to aberrant swimming behaviour, rather than loss of chemoreception. Through investigation of swimming behaviour, we have thus been able for the first time to distinguish between the capacity to respond to an attractant, i.e, chemotaxis, and the capacity to sense an attractant, i.e., chemoreception. Further research into identifying receptors involved in the *Phytophthora* homing response should thus carefully design experiments to distinguish chemotaxis from chemoreception.

The regularity (S6 Fig) and increase (Fig 4D) of tumbling of zoospores of the Gα-deficient *P. infestans* mutant hints to involvement of Gα in tumble suppression. We show that tumbling is the result of flagellar desynchronization (S3 Fig) in line with observations on *P. parasitica* zoospores [30]. For other microswimmers, such as *C. reinhardtii* and sperm cells, it has been shown that flagellar beating is governed by calcium fluxes [51–53], and that reorientation of the flagella is correlated with increased calcium influx rates [54]. As in other eukaryotes, calcium is a key second messenger in oomycetes [55]. There are clues that show calcium to play a role in zoospore chemotaxis and autotaxis [28], and calcium fluxes are associated with encystment [56] and the perception of chemoattractants [57]. Moreover, manipulation of extracellular and internal calcium levels by calcium modulators has shown that calcium plays a role in controlling swimming patterns, with different modulators invoking distinct changes such as perpetual circular or straight swimming, slow spiral 'skidding', or irregular jerky movements [26]. The tumbling increase we observe in the Gα-deficient mutant could well be the result of disturbance of calcium homeostasis, with regular influxes of calcium inducing consistent tumbling. Notability, analyses of *P. sojae* Gα-deficient mutants pointed to Gα as potential negative regulator of intermediates in calcium signalling such as calmodulin and calmodulin-dependent protein kinases [21]. In general evidence for involvement of G-protein signalling components in maintaining calcium oscillations is accumulating [58,59] but to what extent they influence flagellar beating patterns in oomycetes in unknown. Thus, beyond roles in chemoreception, it is worthwhile to investigate the interplay between G-protein mediated signalling and calcium oscillations and influx, and how this affects flagellar beating.

In conclusion, this study demonstrates the power of high speed microscopy to monitor the behaviour of zoospores in real time in the presence of a chemoattractant. By combining high speed microscopy, automatic tracing of zoospore swimming trajectories and intricate quantification of trajectory parameters, we collected a large body of data on zoospore behaviour in an unprecedented manner. By exploring these data and by simulation modelling we were able to show that *Phytophthora* zoospores use positive klinokinesis as a mechanism to remain at the host surface once arriving there, a phase in the homing response coined as retention [14]. Moreover, we showed that our analyses allow us to distinguish between defects in chemotaxis and chemoreception, a prerequisite when identifying candidate receptors for chemoattractants. The methods and insights on zoospore swimming behaviour presented in this study can be instrumental in exploring strategies for *Phytophthora* disease control, for example, for screening efficacy of soil dispersed particle traps laden with (a cocktail of) proven chemoattractants, in 'fishing' out wandering zoospores before arriving at the root, or for identification of chemoreceptor inhibitors or antagonists that disturb the homing response and disable zoospores in finding their targets.

## Materials & methods

### *Phytophthora* strains, culture conditions, and zoosporogenesis

*Phytophthora* strains used in this study are listed in S1 Table. *P. infestans* was maintained on rye sucrose agar (RSA) [60] at 18°C in the dark. Zoosporogenesis was initiated by adding 3 mL

sterile tap water (4˚C) to 10-day-old cultures (60 mm ø petri dish), followed by incubation at 4˚C in the dark for 3h. *P. palmivora* was maintained on 20% V8 agar supplemented with 1 g/l $CaCO_3$ at 25˚C in the light. Zoosporogenesis was initiated by adding 3 mL sterile tap water (4˚C) to 4-day-old cultures (60 mm ø petri dish), followed by incubation at 25˚C in the light for 30 min. *P. capsici* was maintained on 10% V8 agar supplemented with 1 g/l $CaCO_3$ at 25˚C in the light. Zoosporogenesis was initiated by adding 3 mL sterile tap water (4˚C) to 4-day-old cultures (60 mm ø petri dish), followed by incubation at 10˚C in the dark for 30 min, then at 25˚C in the light for 30 min. *P. sojae* was maintained on 10% V8 supplemented with 1 g/l $CaCO_3$ at 25˚C in the light. Zoosporogenesis was initiated by washing 3-day-old cultures (60mm ø petri dish) twice with 3 mL sterile tap water (RT), and adding 3 mL sterile tap water (RT) followed by incubation at 25˚C in the light for 6h. RSA and V8 agar were supplemented with vancomycin (20 μg ml$^{-1}$), ampicillin (100 μg ml$^{-1}$), and amphotericin A (10 μg ml$^{-1}$) to prevent contamination. In addition, for culturing *P. infestans* transformant gs2 the RSA was supplemented with 5 mg ml$^{-1}$ geneticin. The zoospores that were released during incubation were collected by pouring off the tap water. If necessary, the zoospore suspensions were diluted with sterile tap water to a final concentration of $1*10^5$ spores ml$^{-1}$ for experimental use.

### Preparation of amino acid solutions

L-Glutamic acid (Sigma-Aldrich) and L-Phenylalanine (Sigma-Aldrich) were dissolved in Milli-Q water. Stock solutions of Glu and Phe with final concentrations of 100 mM were set to pH 7 with 1.0 M HCl or KOH and diluted with milli-Q water to the desired concentrations for experiments.

### Sample preparation for microscopy

10 μl of zoospore suspension was pipetted onto a microscope slide, gently mixed with 10 μl of water or 10 μl of a Glu or Phe solution, and covered with a cover slip with a 0.2 mm thick ring shaped spacer to create a sealed, circular chamber (Ø 13 mm) allowing unrestricted swimming of zoospores.

### High-speed microscopy

Zoospores were imaged with a Mikatron EoSens CL MC1362 (Unterschleißheim, Germany) high speed camera mounted on a Nikon Eclipse Ti microscope (Tokyo, Japan) equipped with a 10× Plan fluor objective (NA 0.17, DIC). Images were acquired at 250 frames per second for 20 seconds using Epix software (https://www.epixinc.com/products/xcap.htm).

### Data analysis

All data was analysed using MatLab algorithms. Zoospore localization and trajectory linking were performed using the MatLab package of Dufresne and Blair (http://site.physics. georgetown.edu/matlab/index.html), all other analyses were performed using home-written algorithms (MatLab2020b), which are publicly available on the repository Zenodo (https:// zenodo.org/records/11632432). All analyses reported were performed on three independent experiments, each featuring 3*5000 frame image sequences; averages are reported as the mean over these nine replicates, and error bars as the standard deviation between these nine measurements.

Non-motile objects (dust, immotile zoospores, sporangia) were removed from the raw image files by median image filtering. The raw images were then subjected to a Gaussian filter with a width of 3 pixels, to blur the internal structure of zoospore bodies for improving

centroid detection accuracy. Zoospore centroids in the two-dimensional image plane were then localised with sub-pixel resolution and linked together into temporal (xy)-trajectories using a proximity-based search algorithm. To ensure a robust statistical sampling of the swimming behaviour, only trajectories with a length exceeding 2 seconds (i.e. >500 image frames), were used for subsequent analyses.

Instantaneous velocities $v$ at each time point $t$ were determined from the swimming trajectories by computing the Cartesian travel distance in the interval ($t^{-1}$ frames: $t+1$ frames) and dividing by the corresponding time interval. Trajectory curvatures $k$ were determined at each time point by first applying a Savitsky-Golay filter to the spatial path to remove high-frequency positional noise, and subsequently performing a circle-fitting to (xy)-positions at the three time points: ($t-2$ frames: $t$: $t+2$ frames). The curvature was defined as the inverse of the circle radius that best fitted these three temporal positions. Changes in swimming orientation were determined by computing the change in angle ($\theta$) of a trajectory segment ($t$: $t+2$ frames) as compared to the previous segment ($t-2$ frames: $t$), with the horizontal axis as the reference. Reported probability distributions of swimming velocity and curvature were obtained by combining the data for all nine replicates of a given treatment, dose and species.

To quantify the run and tumble phases, we modified the approach used by Tran et al. [30] for zoospores of *P. parasitica*. A tumble phase is associated with a substantial decrease in instantaneous velocity, often approaching 0 μm/s, and a substantial change in the swimming orientation. We identified tumbles as events that satisfy two constraints: (i) the velocity decreases below a threshold value $v_t$ and (ii) the change in swimming direction between the run phase that precedes a tumble and the run phase that follows it, must be larger than a threshold change in angle $\theta_t$. Based on the velocity distributions, we choose $v_t = 50$ μm/s, which for all species is well below the average run velocity. Based on inspections of zoospore trajectories which show undulating run phases associated with small angular variations, we choose $\theta_t = 30$ degrees, such that only real tumbles are identified as such. We have confirmed that the results are robust to variations in these choices. This gives access to the temporal duration of run phases (i.e., the time between two consecutive tumbles), the duration of tumble phases (i.e., the time a zoospore swims at reduced velocity in order to make a turn), and the angular change during a tumble.

Dose-response curves of chemical treatments were constructed in two different ways. A velocity-based dose-response curve is constructed by computing the mean swimming velocity of all entries for a given replicate. Averages and standard deviations are obtained by comparing the results for nine independent replicates. The threshold dose for a given treatment and species was estimated by extrapolating the descending part of the dose-response curves to the velocity for the control sample without chemical treatment. A tumble statistics dose-response curve was constructed by computing the mean duration of run and tumble phases for a given replicate; averages and standard deviations are obtained by comparing the results for nine independent replicates.

Run velocities were determined by fitting the full velocity distribution for a given replicate to a sum of multiple Gaussians (typically three were required for a good fit) and determining the peak position for the Gaussian distribution representing the high-velocity peak. For kinetic fingerprinting of zoospore populations, the mean velocity for a single trajectory, and the logarithm of the mean curvature for that trajectory were used as fingerprint metrics; each dot in these graphs thus represent the mean velocity and mean curvature for a single zoospore trajectory.

To analyse the flagellar beating pattern we make use of the fact the flagellar beating results in a distinct oscillatory pattern in the swimming velocity. We analyse these oscillations using Fourier analysis, where we compute the power-spectral density of the velocity as the squared magnitude of the Fourier transform (using a standard fast-Fourier transform algorithm).

## Simulation model set-up

We simulate non-interacting run-and-tumble swimmers with all input parameters computed or estimated from our experiments on *P. infestans* in homogeneous Glu solutions: a constant run velocity (114 µm s$^{-1}$) with a dose-dependent tumble rate with a perception threshold (computed to be 1.9 mM) and a tumble sequence of deceleration, turning and acceleration, at a native rate of 0.42 tumbles s$^{-1}$. We estimate tumble rates at high Glu concentrations from obtained dose-response curve, which in the model we use to approximate: 0.42 tumbles s$^{-1}$ if [Glu] < threshold and 0.42 * (1 + 0.2 * ([Glu]–threshold)) tumbles s$^{-1}$ if [Glu] > threshold. We assume a new tumble to only start when the previous one is completed. We place a Glu source in the centre of a square simulation box with a concentration gradients based on root exudate profiles described by Fickian diffusion: $c(x) = c_0 \exp(-x_2/\lambda_2)$ where $c_0$ is the source concentration and λ is decay length. λ is based on published data on root exudate profiles [39] and estimated to be between 0.5–2 mm.

## Supporting information

**S1 Fig. Effect of amino acid Phe on trajectories of *P. infestans* zoospores.** Representative trajectories of *P. infestans* zoospores in the absence (0) or presence of Phe at 5 and 50 mM. Curvature ($k$) in 1 µm$^{-1}$, moving direction ($\theta$) in ° and velocity ($v$) in µm s$^{-1}$ are plotted over time (t) in seconds (s).
(TIF)

**S2 Fig. Effects of amino acids Glu and Phe on trajectories of zoospores of four *Phytophthora* species.** Effects of Glu (top row) and Phe (bottom row) on run (left) and tumble (middle) phase duration, and mean velocity (right). Minimum number of zoospores imaged per treatment is 219.
(TIF)

**S3 Fig. Flagellar beating patterns during tumbling.** The stages of tumbling observed in free-swimming *P. infestans* zoospores can be divided into (A) straight swimming, (B) onset of tumble, (C) tumble and (D) recovery. The anterior tinsel flagellum (red arrows), the posterior whiplash flagellum (blue arrows) and the direction of swimming (black arrows) are indicated. Time stamp is from the original time points of the video. Scale bar represents 10 µm.
(TIF)

**S4 Fig. Tumbling angles of *Phytophthora* zoospores.** Tumbling angle probability in the absence (left) and presence of 5 mM (middle) and 50 mM (right) Glu. Minimum number of zoospores analysed is 219.
(TIF)

**S5 Fig. Swimming trajectories of *P. infestans* zoospores in response to phenylalanine (Phe) and glutamic acid (Glu).** Minimum number of zoospores imaged per treatment is 2012. (a) Zoospore trajectories in 5 mM Phe (top row), 50 mM Phe (middle row) and in 50 mM Glu (bottom row) depicted as overlays with the starting point of each trajectory in the centre. Black circles indicate the average effective distance travelled (active diffusion) by zoospores in 1 second. (b) Kinetic fingerprints of swimming zoospores in which each dot depicts the mean velocity for a trajectory of a single zoospore (Y axis) and the logarithm of the mean curvature for that trajectory (X axis) in the absence (0) and presence of Phe in concentrations up to 50 mM. (c) Effect of Phe on run velocity (left), percentage of time spent in tumbles (middle) and active diffusion (right).
(TIF)

**S6 Fig. Analysis of flagellar beating patterns of *P. infestans* zoospores.** (a) Kymograph of tinsel (left) and whiplash (right) flagellum of a *P. infestans* zoospore swimming in a microchamber (middle). (b) Fourier analysis on trajectory velocity fluctuations of zoospores of *P. infestans* wildtype strain 88069 (left) and the *P. infestans* Gα-silenced strain gs2 (right) in response to increasing concentrations glutamic acid (Glu) up to 100 mM, with observed frequencies (Hz) plotted on the Y-axis and power (-) plotted on the X-axis.
(TIF)

**S7 Fig. Simulation model of zoospores reacting to a chemoattractant diffusing from a point source in a square chamber.** Upper panels depict diffusive gradient, lower panels depict spore density after 1 hour of simulated time. Zoospore density given as a time fraction of zoospore presence. (a) Effect of decay length on zoospore accumulation. Point source is 50 mM. (b) Effect of point source concentration on zoospore accumulation.
(TIF)

**S1 Table. *Phytophthora* strains used in this study.**
(PDF)

## Acknowledgments

We thank the Wageningen Light Microscopy Centre (WU) for the use of their facilities, and dr. Martin Lankheet and Remco Pieterse (Laboratory of Experimental Zoology, WU) for technical support and advice.

## Author Contributions

**Conceptualization:** Michiel Kasteel, Joris Sprakel, Tijs Ketelaar, Francine Govers.

**Data curation:** Joris Sprakel, Tijs Ketelaar.

**Formal analysis:** Michiel Kasteel, Joris Sprakel.

**Funding acquisition:** Michiel Kasteel, Tijs Ketelaar, Francine Govers.

**Investigation:** Michiel Kasteel, Tharun P. Rajamuthu.

**Methodology:** Michiel Kasteel, Tharun P. Rajamuthu, Joris Sprakel.

**Project administration:** Tijs Ketelaar, Francine Govers.

**Resources:** Francine Govers.

**Software:** Joris Sprakel.

**Supervision:** Joris Sprakel, Tijs Ketelaar, Francine Govers.

**Validation:** Michiel Kasteel, Joris Sprakel.

**Visualization:** Michiel Kasteel, Joris Sprakel.

**Writing – original draft:** Michiel Kasteel, Tijs Ketelaar, Francine Govers.

**Writing – review & editing:** Michiel Kasteel, Tijs Ketelaar, Francine Govers.

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
