## [Decision Letter · Decision Letter 0]

12 Aug 2024

Dear MSc Kasteel,

Thank you very much for submitting your manuscript "*Phytophthora* zoospores display klinokinetic behaviour in response to a chemoattractant" for consideration at PLOS Pathogens. As with all papers reviewed by the journal, your manuscript was reviewed by members of the editorial board and by several independent reviewers. In light of the reviews (below this email), we would like to invite the resubmission of a significantly-revised version that takes into account the reviewers' comments.

*We cannot make any decision about publication until we have seen the revised manuscript and your response to the reviewers' comments. Your revised manuscript is also likely to be sent to reviewers for further evaluation.*

*When you are ready to resubmit, please upload the following:*

*[1] A letter containing a detailed list of your responses to the review comments and a description of the changes you have made in the manuscript. Please note while forming your response, if your article is accepted, you may have the opportunity to make the peer review history publicly available. The record will include editor decision letters (with reviews) and your responses to reviewer comments. If eligible, we will contact you to opt in or out.*

*[2] Two versions of the revised manuscript: one with either highlights or tracked changes denoting where the text has been changed; the other a clean version (uploaded as the manuscript file).*

*Important additional instructions are given below your reviewer comments.*

*Please prepare and submit your revised manuscript within 60 days. If you anticipate any delay, please let us know the expected resubmission date by replying to this email. Please note that revised manuscripts received after the 60-day due date may require evaluation and peer review similar to newly submitted manuscripts.*

*Thank you again for your submission. We hope that our editorial process has been constructive so far, and we welcome your feedback at any time. Please don't hesitate to contact us if you have any questions or comments.*

*Sincerely,*

Yuanchao Wang

Academic Editor

PLOS Pathogens

Bart Thomma

Section Editor

PLOS Pathogens

Michael Malim

Editor-in-Chief

PLOS Pathogens

orcid.org/0000-0002-7699-2064

*Reviewer's Responses to Questions*

*
**Part I - Summary**
*

*Please use this section to discuss strengths/weaknesses of study, novelty/significance, general execution and scholarship.*

*Reviewer #1: The molecular basis of chemoattraction of unicellular eukaryotes is certainly of topic of potential interest to a wider audience.*

*Reviewer #2: The manuscript provides valuable insights into the chemotactic responses of zoospores of the genus Phytophthora. The use of high-speed cameras to track swimming trajectories and quantify key parameters is a commendable approach that has yielded significant findings. Furthermore, the manuscript distinguishes between chemotaxis deficiencies and chemoreception impairments by analysing the Gα-deficient mutants.*

*The manuscript is well written and the results are presented in a clear and concise manner. Nevertheless, the following comments should be addressed to improve the clarity of the manuscript:*

*Reviewer #3: Zoospores is asexual spores of oomycetes and play important roles for the dispersal of these pathogens. These two flagellated single cells swim toward host through chemotaxis mechanism, which is not fully understand until now. The manuscript entitled “Phytophthora zoospores display klinokinetic behaviour in response to a chemoattractant” monitored swimming trajectories of four Phytophthora species and described swimming behaviour with quantitative parameters such as velocity v, curvature K and moving direction θ. With these parameters, the difference of swimming and chemotaxis behaviour of four Phytophthora species were compared. Moreover, a P. infestans Gα mutant was used to compare swimming behaviour and chemotaxis. The manuscript provided another avenue to understand swimming and chemotaxis behaviour of Phytophthora species, which are suitable for publication in Plos Pathogens.*

*
**Part II – Major Issues: Key Experiments Required for Acceptance**
*

*Generally, there should be no more than 3 such required experiments or major modifications for a "Major Revision" recommendation. If more than 3 experiments are necessary to validate the study conclusions, then you are encouraged to recommend "Reject".*

*Reviewer #1: The present study does not take into account some very old literature about the role of calcium and its affect on the frequency of turning. I surmise that a calcium influx mediated via a G coupled receptor almost certainly provides the central intracellular signal to coordinate the same response from different types of chemoreceptors.*

Reid, B., Morris, B. W., and Gow, N. A. R. 1995. Calcium-dependent,

genus specific, autoaggregation of zoospores of phytopathogenic fungi.

Exp. Mycol. 19: 202–213.

Donaldson, S. P., and Deacon, J.W. 1993. Changes in motility of Pythium

zoospores induced by calcium and calcium-modulating drugs. Mycol.

Res. 97: 877–883.

Connolly MS, Williams N, Heckman CA, Morris PF. Soybean isoflavones trigger a calcium influx in Phytophthora sojae. Fungal Genet Biol. 1999 Oct;28(1):6-11. doi: 10.1006/fgbi.1999.1148. PMID: 10512667.

Xu, C., & Morris, P. F. (1998). External calcium controls the developmental strategy of Phytophthora sojae cysts. Mycologia, 90(2), 269–275. 10.1080/00275514.1998.12026906

*I too have noted that 1mM Ca promotes very straight swimming in response to a chemotactic signal.*

*Reviewer #2: (No Response)*

*Reviewer #3: No modifications are required.*

*
**Part III – Minor Issues: Editorial and Data Presentation Modifications**
*

*Please use this section for editorial suggestions as well as relatively minor modifications of existing data that would enhance clarity.*

*Reviewer #1: (No Response)*

*Reviewer #2: 1 Could the authors provide a rationale for not including L-Phenylalanine as a control in the experiment analysing zoospore movement of P. infestans gs2? The inclusion of L-phenylalanine as an additional control could help to further validate that the ability of gs2 to respond to chemotaxis was not affected.*

2 In the Results section, lines 153 and 174-177, the manuscript refers to figures as Fig. S2a, b, c. However, the sub-figures do not have the corresponding letter annotations in Fig. S2, and there is no description of the letter annotations in the figure legend.

3. On lines 301 and 303, the manuscript refers to Fig. 3d, but this figure is not included in the submitted material. Could this be a case of mislabelling, and did you intend to refer to Fig. 4d instead?

*4 Line 270, the second half of the sentence beginning with "and why" seems disjointed and unclear in its current form. It would benefit from rephrasing to ensure that the reason for oomycete zoospores' tendency to tumble upon collision with a surface is explained in a way that is logically connected to the first part of the sentence.*

*Reviewer #3: No modifications are required.*

*PLOS authors have the option to publish the peer review history of their article (what does this mean?). If published, this will include your full peer review and any attached files.*

**

*Reviewer #1: **Yes: **Paul F. Morris*

*Reviewer #2: No*

*Reviewer #3: No*

* *

*
Figure Files:
*

*While revising your submission, please upload your figure files to the Preflight Analysis and Conversion Engine (PACE) digital diagnostic tool, https://pacev2.apexcovantage.com. PACE helps ensure that figures meet PLOS requirements. To use PACE, you must first register as a user. Then, login and navigate to the UPLOAD tab, where you will find detailed instructions on how to use the tool. If you encounter any issues or have any questions when using PACE, please email us at figures@plos.org.*

*
Data Requirements:
*

*Please note that, as a condition of publication, PLOS' data policy requires that you make available all data used to draw the conclusions outlined in your manuscript. Data must be deposited in an appropriate repository, included within the body of the manuscript, or uploaded as supporting information. This includes all numerical values that were used to generate graphs, histograms etc.. For an example see here on PLOS Biology: http://www.plosbiology.org/article/info%3Adoi%2F10.1371%2Fjournal.pbio.1001908#s5.*

*
Reproducibility:
*

*To enhance the reproducibility of your results, we recommend that you deposit your laboratory protocols in protocols.io, where a protocol can be assigned its own identifier (DOI) such that it can be cited independently in the future. Additionally, PLOS ONE offers an option to publish peer-reviewed clinical study protocols. Read more information on sharing protocols at https://plos.org/protocols?utm_medium=editorial-email&utm_source=authorletters&utm_campaign=protocols*

---

## [Editor Report · Decision Letter 1]

10 Sep 2024

Dear MSc Kasteel,

We are pleased to inform you that your manuscript '*Phytophthora* zoospores display klinokinetic behaviour in response to a chemoattractant' has been provisionally accepted for publication in PLOS Pathogens.

*Before your manuscript can be formally accepted you will need to complete some formatting changes, which you will receive in a follow up email. A member of our team will be in touch with a set of requests.*

*Please note that your manuscript will not be scheduled for publication until you have made the required changes, so a swift response is appreciated.*

*IMPORTANT: The editorial review process is now complete. PLOS will only permit corrections to spelling, formatting or significant scientific errors from this point onwards. Requests for major changes, or any which affect the scientific understanding of your work, will cause delays to the publication date of your manuscript.*

*Should you, your institution's press office or the journal office choose to press release your paper, you will automatically be opted out of early publication. We ask that you notify us now if you or your institution is planning to press release the article. All press must be co-ordinated with PLOS.*

*Thank you again for supporting Open Access publishing; we are looking forward to publishing your work in PLOS Pathogens.*

*Best regards,*

Yuanchao Wang

Academic Editor

PLOS Pathogens

Bart Thomma

Section Editor

PLOS Pathogens

Michael Malim

Editor-in-Chief

PLOS Pathogens

orcid.org/0000-0002-7699-2064
---

## [Editor Report · Acceptance letter]

25 Sep 2024

Dear MSc Kasteel,

We are delighted to inform you that your manuscript, "Phytophthora zoospores display klinokinetic behaviour in response to a chemoattractant," has been formally accepted for publication in PLOS Pathogens.

Best regards,

Michael Malim

Editor-in-Chief

PLOS Pathogens

orcid.org/0000-0002-7699-2064